# *MADFormer*: Mixed Autoregressive and Diffusion Transformers for Continuous Image Generation

**Junhao Chen**
Tsinghua University
`chenjunh22@mails.tsinghua.edu.cn`

**Yulia Tsvetkov**
University of Washington
`yuliats@cs.washington.edu`

**Xiaochuang Han**
University of Washington
`xhan77@cs.washington.edu`

## Abstract

Recent progress in multimodal generation has increasingly combined autoregressive (AR) and diffusion-based approaches, leveraging their complementary strengths: AR models capture long-range dependencies and produce fluent, context-aware outputs, while diffusion models operate in continuous latent spaces to refine high-fidelity visual details. However, existing hybrids often lack systematic guidance on how and why to allocate model capacity between these paradigms. In this work, we introduce *MADFormer*, a Mixed Autoregressive and Diffusion Transformer that serves as a testbed for analyzing AR-diffusion trade-offs. *MADFormer* partitions image generation into spatial blocks, using AR layers for one-pass global conditioning across blocks and diffusion layers for iterative local refinement within each block. Through controlled experiments on FFHQ-1024 and ImageNet, we identify two key insights: (1) block-wise partitioning significantly improves performance on high-resolution images, and (2) vertically mixing AR and diffusion layers yields better quality-efficiency balances—improving FID by up to 75% under constrained inference compute. Our findings offer practical design principles for future hybrid generative models. Code and models will be released upon publication.

## 1 Introduction

Multimodal generative models have achieved remarkable success in vision-language tasks such as text-to-image generation, image captioning, and video synthesis (Saharia et al., 2022; Luo et al., 2022; Han et al., 2022; Zhang et al., 2024). These systems typically adopt autoregressive (AR) models for language, generating discrete tokens one at a time with strong contextual coherence, and diffusion models for image generation, operating in the continuous domain by progressively denoising latent representations. This division leverages the strengths of both: AR excels at structured sequence modeling, while diffusion achieves high image fidelity through iterative refinement.

For the image generation component, current approaches follow three main paradigms. One line of work uses AR models on discrete visual tokens, enabling reuse of large language model architectures but often introducing quantization artifacts and limiting generation quality (Esser et al., 2020; Team, 2024; Agarwal et al., 2025). Another line adopts fully diffusion-based models in continuous latent spaces, producing high-quality images with global coherence but at the cost of slow sampling and high computational overhead (Rombach et al., 2021; Peebles & Xie, 2022; Esser et al., 2024; Labs, 2025). A third line of research explores hybrid architectures that combine AR and diffusion generation within the image processing pathway, aiming to integrate the efficiency and structure of AR with the fine-grained detail refinement of diffusion (Li et al., 2024; Hu et al., 2024). These early efforts point to the promise of unifying the two paradigms, but the design space remains underexplored.

In this work, we present *MADFormer*, a unified Transformer-based architecture for continuous image generation that integrates AR and diffusion modeling across two axes—token sequences and model

layers.[1] Apart from introducing a novel approach, *MADFormer*, more importantly, serves as a testbed for systematically exploring the hybrid design space and deriving generalizable insights. We build on a vanilla architecture that employs AR for language and diffusion for image generation, extending it with intra-image AR conditioning and flexible allocation of AR and diffusion blocks. By varying structure and capacity distribution between these components, we aim to better understand their interaction and provide actionable guidance for designing efficient, high-quality multimodal models.

**Mixing AR and diffusion across tokens.** Along the token axis, we partition the image into coarse-grained blocks (e.g., 16 blocks of $256 \times 256$ patches for an image with $1024 \times 1024$ latent patches). AR modeling operates across blocks: generation of each block is conditioned autoregressively on all previous ones. Within each block, we apply a diffusion objective to iteratively denoise continuous latents, overall balancing structure and flexibility.

**Mixing AR and diffusion across layers.** Along the depth axis, we split the Transformer into AR and diffusion layers. Early layers run autoregressively, processing previously generated blocks to compute a prior for the next block. Late layers follow the diffusion objective: conditioned on both the noisy block and the AR output, they denoise towards the ground truth. Importantly, we maintain a unified architecture—only the training objectives differ. This hybrid layout is especially effective under constrained compute: the AR path provides a strong initialization, enabling the diffusion layers to converge with fewer iterations.

We validate our approach on FFHQ-1024 and ImageNet, with experiments designed to isolate the effects of image partitioning, model depth allocation, inference budget, etc. Our contributions are:

1. We propose *MADFormer*, a unified Transformer architecture that supports flexible autoregressive and diffusion integration within the image generation tower. Our design preserves architectural simplicity while enabling structured exploration of AR-diffusion trade-offs.

2. We conduct a study across design axes—diffusion depth, AR block granularity, auxiliary modules, and loss designs—deriving empirical insights into AR-diffusion interactions.

3. We derive practical guidelines for allocating model capacity between AR and diffusion components, showing that AR-heavy models are advantageous under budget constraints, while diffusion-heavy models achieve higher fidelity when compute allows. We find that increasing AR layer allocation can improve FID scores by 60-75% under constrained inference compute.

## 2 PREREQUISITES

### 2.1 AUTOREGRESSIVE MODELING

AR models decompose the joint probability of a sequence $x = (x_1, x_2, \ldots, x_n)$ as $p(x) = \prod_{i=1}^{n} p(x_i \mid x_{1:i-1})$. A Transformer with causal masking estimates each conditional by computing a hidden state $z_i = f(x_{1:i-1}), \quad p(x_i \mid x_{1:i-1}) = p(x_i \mid z_i)$. Training minimizes the negative log-likelihood, implemented as cross-entropy for discrete tokens or continuous density loss for real-valued inputs. Recent work shows that AR modeling can be extended as well to the continuous latent space (Li et al., 2024).

### 2.2 DIFFUSION

Diffusion models typically learn to reverse a noise process in the continuous space. Starting with clean latents $x_0$, the forward process adds Gaussian noise via $q(x_t \mid x_{t-1}) = \mathcal{N}(x_t; \sqrt{1 - \beta_t}\, x_{t-1}, \beta_t I)$, with direct sampling $x_t = \sqrt{\bar{\alpha}_t}\, x_0 + \sqrt{1 - \bar{\alpha}_t}\, \varepsilon, \quad \varepsilon \sim \mathcal{N}(0, I)$, where $\bar{\alpha}_t = \prod_{s=1}^{t}(1 - \beta_s)$. The reverse process uses a network $\varepsilon_\theta(x_t, t, z)$, conditioned on optional input $z$, to model $p_\theta(x_{t-1} \mid x_t) = \mathcal{N}(x_{t-1}; \mu_\theta(x_t, t, z), \sigma_t^2 I)$. Training minimizes noise prediction error: $\mathbb{E}_{t,x_0,\varepsilon} \|\varepsilon_\theta(x_t, t, z) - \varepsilon\|^2$. At inference, the model denoises $x_T \sim \mathcal{N}(0, I)$ into $x_0$ (Ho et al., 2020).

---

[1]Our integration of AR and diffusion focuses on mixing within the image generation process itself, rather than across modalities as in unified models like Transfusion (Zhou et al., 2024) or Show-o (Xie et al., 2024).

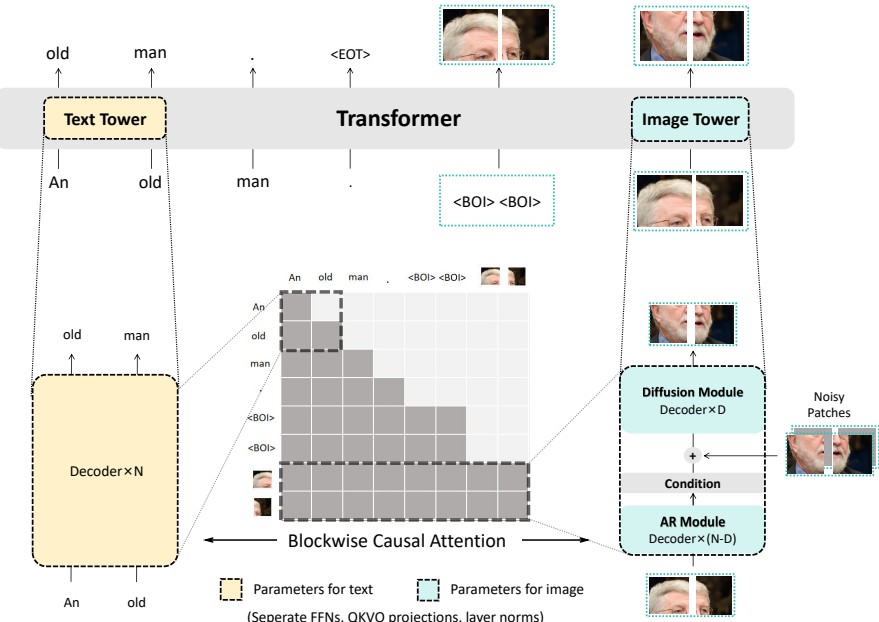

Figure 1: **High-level overview of the *MADFormer* architecture.** A single Transformer processes all modalities as a unified sequence. Text tokens follow a next-token prediction objective, while image tokens are grouped into blocks trained autoregressively with a diffusion objective. We use separate parameters (FFNs, QKVO projections, and layer norms) for each modality. The Transformer is divided into two stages: early layers produce conditions from text and image blocks; later layers denoise noisy image blocks. Each block attends to itself and preceding clean blocks.

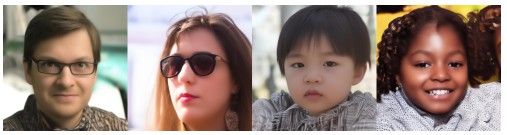

(a) **Samples on FFHQ** 1024×1024 **with 9 inference steps.** Despite minor imperfections in backgrounds and details, *MADFormer* generates coherent high-res human faces with strong structural consistency in minimal steps.

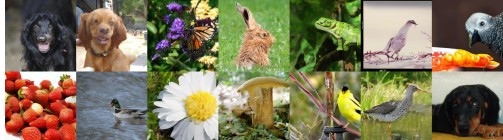

(b) **Samples on ImageNet** $256 \times 256$. *MADFormer* demonstrates understanding of class prompts and generates well-structured images, though fine details can still be improved with more training.

Figure 2: **Qualitative results of *MADFormer* with 14 diffusion layers (out of 28 total).** The AR conditioning mechanism enables strong sample quality even under limited inference steps. More qualitative results can be found in Appendix C.

**Continuous Image Tokenizers**    In image generation, we operate on continuous VAE latents rather than discrete tokens, leveraging more mature and effective tokenizers without quantization in the latent space (Rombach et al., 2021; Agarwal et al., 2025). This choice simplifies AR factorization and diffusion scheduling while preserving high-fidelity visual detail. Continuous latents allow direct modeling with Gaussian noise and real-valued predictions, avoiding discrete codebooks and related artifacts. (van den Oord et al., 2017; Esser et al., 2020; Mentzer et al., 2023; Yu et al., 2023; Tschannen et al., 2023)

# 3 *MADFormer*: A TESTBED FOR ANALYZING AR-DIFFUSION DESIGN TRADE-OFFS

## 3.1 FRAMEWORK

We introduce *MADFormer*, a unified Transformer architecture designed not only as a standalone method but as a testbed for analyzing architectural trade-offs between AR and diffusion modeling in continuous image generation. Built on top of recent advances in hybrid generation (Li et al., 2024; Zhou et al., 2024; Shi et al., 2024; Hu et al., 2024; Peebles & Xie, 2022), *MADFormer* integrates key components from prior work into a cohesive and extensible framework. Its modular design enables systematic exploration of how to balance global sequence modeling via AR conditioning and fine-grained detail refinement via diffusion across both token sequences and network depth.

**Data Representation.** We preprocess each modality according to its characteristics. Text inputs are tokenized into discrete tokens by the Llama 3 tokenizer (Dubey et al., 2024). Images are mapped into a continuous latent space via the Stable Diffusion variational autoencoder (VAE) (Rombach et al., 2021). We linearize the image latents in left-to-right, top-to-bottom order, delimit them with BOI (beginning-of-image) and EOI (end-of-image) tokens, and interleave them with text tokens in accordance with the original sequence. This produces a unified sequence of discrete text tokens and continuous image latents. Image latent patches are grouped into contiguous blocks with bidirectional attention within, and the blocks are treated as tokens in the autoregressive sequence.

**Model Pipeline.** Our architecture primarily consists of a single Transformer built from standard Llama decoder blocks, enabling unified AR modeling and diffusion-based generation in latent space. To project different modalities into this space, we employ modality-specific components: text tokens are embedded via learned embedding matrices, while image latent patches are processed through U-Net (Nichol & Dhariwal, 2021; Saharia et al., 2022) upsampling and downsampling blocks. Unlike the DiT paradigm (Peebles & Xie, 2022), we do not inject explicit conditioning (e.g., timesteps) into individual Transformer layers. Instead, timestep information is encoded directly into the image latents via the U-Net downsampler.

We conceptually divide the Transformer's decoder stack into two stages: the early layers serve as an AR conditioning module, while the later layers execute recursive diffusion denoising. At each diffusion step, the condition generated by the early layers is added to the noisy latent prior to the denoising loop. The model predicts the denoised latent from the current noisy latent $x_t$, following a timestep schedule.

In addition, we adopt the strategy from ACDiT (Hu et al., 2024) of prepending clean image blocks to the noisy ones. This enhances contextual information for AR modeling and improves denoising fidelity, though it incurs additional computational cost. All modalities share the Transformer backbone but use separate parameter sets for processing: text, clean image blocks, and noisy image blocks—referred to as the text tower, clean tower, and noise tower, respectively (Liang et al., 2024). Cross-modal interaction is facilitated via interleaved causal attention, with bidirectional attention applied within image blocks. To support mixed clean and noisy inputs, attention masks are dynamically constructed using Flex Attention (Dong et al., 2024). We demonstrate the effectiveness of this architectural design through ablations in Sec. 4.4.

## 3.2 DESIGN SPACE

We explore three primary axes in the design space of our model: diffusion depth, AR length, and loss function. These components fundamentally affect training convergence, generation speed, and output quality, independent of model size.

**Diffusion Depth.** We define *diffusion depth* as the number of Transformer layers dedicated to the denoising process, corresponding to the $D$ decoder blocks shown in the *Image Tower* of Figure 1. While prior work often employs the entire model for diffusion, we argue that this results in unnecessary computational overhead. We posit that a single-pass AR conditioning stage can effectively capture both inter-modality and inter-block dependencies, allowing only a small subset of later layers to refine intra-block details through recursive denoising. In Section 4.2, we compare using the full model, $\frac{3}{4}$,

$\frac{1}{2}$, and $\frac{1}{4}$ of the layers for diffusion, and in Section 4.1, we propose an optimal AR/diffusion layer ratio under constrained inference budgets.

We formalize this layer-wise structure by formulation: the image tower consists of layers with the first $N-D$ allocated for AR conditioning and the final $D$ for diffusion-based denoising. The AR stage computes a conditioning representation over previously generated image blocks:

$$\mathbf{h}_0 = \texttt{Embed}(\mathbf{z}_{\text{prev}}) + \texttt{PosEnc}, \tag{1}$$

$$\mathbf{h}_i = \texttt{DecoderBlock}_i(\mathbf{h}_{i-1}), \quad i = 1, \dots, N-D, \tag{2}$$

$$\mathbf{z}_{\text{cond}} = \mathbf{h}_{N-D}. \tag{3}$$

Here, $\texttt{PosEnc}$ is a shorthand for the multidimensional rotary position encoding (RoPE-ND) as proposed in ACDiT (Hu et al., 2024). To begin the diffusion stage, we inject the noised ground-truth latent, $\sqrt{\bar{\alpha}_t}\,\mathbf{z}_{\text{image}} + \sqrt{1 - \bar{\alpha}_t}\,\boldsymbol{\epsilon}$, where $\boldsymbol{\epsilon}$ is standard Gaussian noise, along with the condition $\mathbf{z}_{\text{cond}}$. The diffusion layers aim to denoise $\sqrt{\bar{\alpha}_t}\,\mathbf{z}_{\text{image}} + \sqrt{1 - \bar{\alpha}_t}\,\boldsymbol{\epsilon}$ into the clean latent $\mathbf{z}_{\text{image}}$[2]:

$$\mathbf{h}_{N-D} = \sqrt{\bar{\alpha}_t}\,\mathbf{z}_{\text{image}} + \sqrt{1 - \bar{\alpha}_t}\,\boldsymbol{\epsilon} + \mathbf{z}_{\text{cond}}, \tag{4}$$

$$\mathbf{h}_{N-D+j} = \texttt{DecoderBlock}_{N-D+j}(\mathbf{h}_{N-D+j-1}), \quad j = 1, \dots, D. \tag{5}$$

The final output $\hat{\mathbf{z}}_{\text{image}} = \texttt{Proj}(\mathbf{h}_N)$ serves as the model's prediction of the clean latent.

**AR Length.** We study the optimal granularity for autoregressive processing of image patches. AR length refers to the number of image blocks to which an input image is partitioned (e.g., AR length is 2 in Figure 1's demonstration). Different partitioning strategies significantly affect the quality of AR condition modeling while incurring minimal computational overhead. In Section 4.2, we evaluate splits of 4, 16, and 64 blocks under varying image resolutions, and propose a recommended splitting strategy based on empirical results.

**Loss Function.** Prior work has emphasized the importance of loss function design for stable training and high-quality generation. In line with common practice, we apply negative log-likelihood loss on text tokens and mean squared error on image latent predictions. Additionally, our architecture enables two new auxiliary loss terms, each modulated by a tunable hyperparameter. We report their effects in Section 4.6.

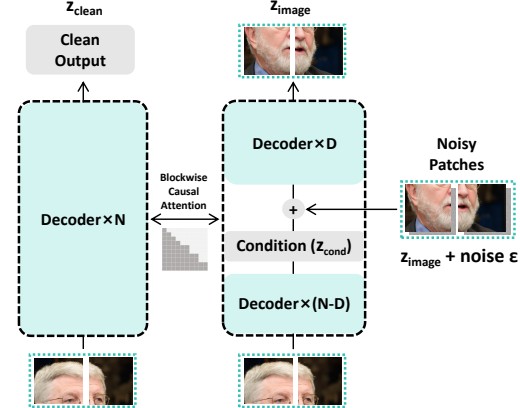

- **Hidden Loss.** To encourage the AR module to produce conditions informative for denoising, we introduce a loss between the AR-generated condition and the ground truth latent of the next image block. Ideally, the condition should fully encode the clean latent it precedes.

- **Clean Tower Loss.** We apply an auxiliary loss between the output of each clean image block and its corresponding next clean latent block, analogous to next-token prediction in AR models.

Figure 3: **Architectural details of *MADFormer*'s image tower.** The hidden loss and clean tower loss are computed using the clean tower output and conditioning input, compared against the ground truth image block.

Together with gradients backpropagated via attention involving noisy blocks, this objective encourages the clean tower to encode predictive signals beneficial for denoising.

The total loss is computed as a weighted sum of the individual objectives:

$$\mathcal{L}_{\text{total}} = \lambda_{\text{text}} \cdot (-\log p(\mathbf{y}_{\text{text}} \mid \mathbf{x})) + \lambda_{\text{image}} \cdot \|\hat{\mathbf{z}}_{\text{image}} - \mathbf{z}_{\text{image}}\|_2^2 +$$

$$\lambda_{\text{hidden}} \cdot \|\mathbf{z}_{\text{condition}} - \mathbf{z}_{\text{image}}\|_2^2 + \lambda_{\text{tower}} \cdot \|\mathbf{z}_{\text{clean}} - \mathbf{z}_{\text{image}}\|_2^2, \tag{6}$$

---

[2]Future work may consider alternative parameterizations such as predicting $\boldsymbol{\epsilon}$ (Ho et al., 2020) or the velocity vector (Salimans & Ho, 2022).

where $\mathbf{z}_{\text{condition}}$ and $\mathbf{z}_{\text{image}}$ are as demonstrated in Figure 3 . $\lambda_{\text{text}} = 1$ and $\lambda_{\text{image}} = 5$ are fixed weights, following Transfusion (Zhou et al., 2024); $\lambda_{\text{hidden}}$ and $\lambda_{\text{tower}}$ are tunable hyperparameters analyzed in Section 4.6.

### 3.3 EXPERIMENT SETUP

Through a suite of experiments and evaluations, we demonstrate the breadth of our model's design space and present key insights derived from comprehensive ablation studies.

**Datasets**  We conduct ablation studies on two widely used datasets. The FFHQ dataset (Karras et al., 2018) contains 70,000 high-quality human face images at a resolution of $1024 \times 1024$. The ImageNet dataset (Russakovsky et al., 2015) includes approximately 1.28 million images across 1,000 classes at a resolution of $256 \times 256$.

**Training**  For experiments on FFHQ-1024, we use a 1.3B parameter model with 28 decoder layers, consisting of two parameter sets for processing clean and noised image blocks. For ImageNet abla-tions, we adopt a 2.1B parameter model, which includes additional components for text processing: a text tower, token embeddings, and a language modeling head. In both configurations, the U-Net upsampler and downsampler contribute approximately 0.2B parameters. The VAE is kept frozen during training; only the rest of the model is updated.

We train using the AdamW optimizer with a Warmup Stable Decay (WSD) learning rate schedule, a peak learning rate of $3 \times 10^{-4}$, and a weight decay of $5 \times 10^{-2}$. An exponential moving average (EMA) with a decay factor of $0.9999$ is applied to stabilize training. For image denoising during training, we use $1,000$ diffusion steps, following the DDPM timestep scheduler (Ho et al., 2020).

On FFHQ-1024, we train for $210,000$ steps with a batch size of $64$. On ImageNet, we train for $250,000$ steps with a batch size of $256$. Additional training configurations and architectural details are provided in Appendix B. Note that the relatively high FID (Fréchet Inception Distance) scores in our ablations for the ImageNet dataset, compared to MAR and ACDiT, are primarily due to significantly fewer training epochs (50 vs. 400 for MAR and 800 for ACDiT) and the omission of classifier-free guidance (CFG), as our focus is on controlled design-space analysis rather than final performance.

**Evaluation**  We use Fréchet Inception Distance (FID) (Heusel et al., 2017) as the primary metric for image quality. For FFHQ-1024, FID is computed over $8,000$ samples; for ImageNet, we follow the standard FID-50K protocol with $50,000$ generated images. The samples are generated using the DDIM sampler (Song et al., 2020), with $250$ steps for FFHQ-1024 and $100$ for ImageNet, *unless otherwise noted in ablations*[3]. Final FID scores are averaged over the last five checkpoints (every $10,000$ steps) to reduce variance. We also report the number of function evaluations (NFE) to compare inference speed and compute efficiency across configurations.

## 4 RESULTS

Using our *MADFormer* testbed, we systematically explore the design space of hybrid AR-diffusion models along two main axes: (1) the *layer axis*, which controls the division of Transformer layers between AR and diffusion, and (2) the *token axis*, which determines how image latents are segmented autoregressively. These axes govern the trade-off between generation quality and computational efficiency. We also present auxiliary ablations on architectural variants and training strategies, providing practical guidance for building efficient, high-fidelity AR-diffusion generators.

---

[3]Advanced ODE samplers (e.g., DPM-Solver/++ (Lu et al., 2022a;b)) can lower the effective NFE for any model, but they are orthogonal to our AR-diffusion capacity allocation question; we therefore keep standard settings to isolate architectural effects.

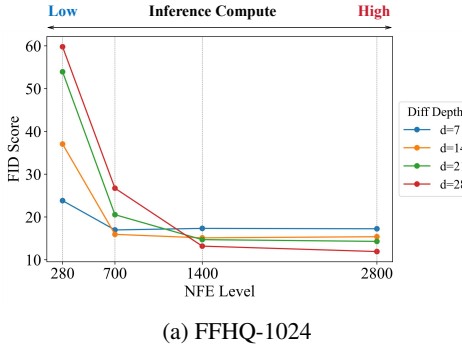
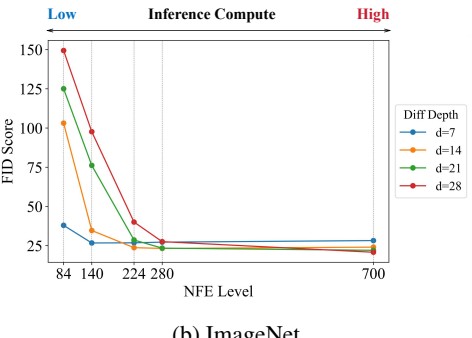

(a) FFHQ-1024          (b) ImageNet

Figure 4: **FID vs. NFE for different AR-Diffusion layer ratios.** Under low NFE (i.e., constrained compute), AR-heavy configurations (e.g., 3:1 AR:Diffusion with $d = 7$) consistently outperform diffusion-heavy ones (e.g., all diffusion layers with $d = 28$), decreasing FID by up to 60-75%. As compute increases, diffusion-heavy setups yield better performance, demonstrating the complementary roles of AR for structure and diffusion for refinement.

## 4.1 LAYER AXIS: AR-DIFFUSION ALLOCATION

> **TOP TAKEAWAY**
>
> **When inference budget is limited, the optimal AR–diffusion layer allocation prioritizes AR over diffusion modeling (Figure 4).**

This emerges as a key insight from our ablation studies. We evaluate model performance under varying compute budgets, measured by the NFE during inference—specifically 280, 700, 1400, and 2800 for FFHQ, and 84, 140, 224, 280 and 700 for ImageNet. For each setting, we compare models with different AR-diffusion layer splits within a fixed 28-layer budget. As shown in Figure 4, allocating more layers to AR consistently improves FID scores in low-NFE settings. On both datasets, AR-heavy configurations outperform diffusion-heavy ones by a wide margin—achieving up to 60–75% FID improvement under tight compute constraints. However, as compute increases, the trend reverses: diffusion-heavy configurations begin to dominate, particularly on high-resolution datasets. These results indicate that AR layers are more compute-efficient for modeling global structure, while diffusion excels at fine-grained refinement when budget allows.

**Increasing diffusion depth improves generation fidelity.** We further ablate the number of layers allocated to diffusion while keeping the total depth fixed at 28 and the same number of diffusion steps.[4] As shown in Table 1, increasing diffusion depth improves FID consistently. This suggests that both AR and diffusion components benefit from substantial capacity, and highlights the importance of judiciously balancing these components depending on the generation setting.

Table 1: **Ablation on diffusion depth.** All models in our experiments are trained for and 210k steps on FFHQ, and 250k steps (50 epochs) on ImageNet.

|  | Diffusion Depth | | | |
|---|---|---|---|---|
| **FID** ($\downarrow$) | $d = 7$ | $d = 14$ | $d = 21$ | $d = 28$ |
| **FFHQ** | 20.2 | 17.8 | 16.6 | **15.9** |
| **ImageNet** | 34.0 | 30.0 | 28.1 | **27.4** |

## 4.2 TOKEN AXIS: AR BLOCK GRANULARITY

**Optimal AR length depends on image resolution.** We define *AR length* as the number of image blocks processed sequentially, determined by how the image is partitioned for AR modeling. For example, an FFHQ image at $1024 \times 1024$ resolution with $l = 4$ is divided into four $512 \times 512$

---

[4]Since the diffusion process steps through more layers with diffusion-heavy models than AR-heavy models, the NFE in the setup for diffusion-heavy models is higher, a different setup from Figure 4.

patches. As shown in Table 2, the optimal AR length varies by dataset: FFHQ performs best with 16 blocks of $256 \times 256$, while ImageNet prefers a single block of the same size. These findings align with ACDiT (Hu et al., 2024) on ImageNet, where longer AR sequences degrade quality. However, our FFHQ-1024 experiments reveal a more nuanced trend, suggesting that higher-resolution images benefit from finer-grained AR decomposition. We argue that the optimal AR length depends on image resolution, architecture, and dataset characteristics, highlighting the need for future work in these directions.

Table 3: Ablation on clean blocks and AR condition

| FID ($\downarrow$) | Clean Blocks + Condition | Clean Blocks Only | Condition Only |
|---|---|---|---|
| **FFHQ** | **17.8** | 20.1 | 19.7 |
| **ImageNet** | **30.0** | 31.9 | 31.2 |

### 4.3 AUXILIARY MODULES FOR ENHANCING DIFFUSION

**Clean blocks and AR conditioning both enhance diffusion quality.** We introduce two auxiliary modules to guide the diffusion process: (1) clean blocks—uncorrupted image representations prepended to the input sequence—and (2) AR conditioning—autoregressively generated context from previous blocks. These inject structure into the denoising trajectory and serve as complementary priors. As shown in Table 3, ablating either module leads to consistent FID degradation on FFHQ and ImageNet, confirming their individual effectiveness and the benefit of combining them.

Table 2: Ablation on AR length

| FID ($\downarrow$) | AR Length | | | |
|---|---|---|---|---|
| | $l=1$ | $l=4$ | $l=16$ | $l=64$ |
| **FFHQ** | - | 18.9 | **17.8** | 21.9 |
| **ImageNet** | **28.4** | 30.0 | 34.5 | 39.7 |

### 4.4 MODALITY-SPECIFIC SEPARATION

**Using separate parameter sets for text, clean, and noisy image blocks has a trivial effect in our experiments.** We explored introducing sparsity into the model by assigning distinct parameter sets (e.g., FFNs, QKVO projections, and layer norms) to text, clean and noisy image blocks. This

Table 4: Ablation on param sets

| FID ($\downarrow$) | Full Design | Single Set |
|---|---|---|
| **FFHQ** | **17.8** | 17.8 |
| **ImageNet** | **30.0** | 30.4 |

ablation on model design is motivated by the intuition that separating parameter spaces could facilitate learning distinct distributions across modalities, an idea supported by LMFusion (Shi et al., 2024). However, our ablation results in Table 4 show a weak improvement in performance. In other words, a dense model with shared parameters across all modalities remains effective.

### 4.5 MLP-STYLE DENOISING

Table 5: Ablation on MLP denoising

**Sequence-level causal attention during diffusion is essential.** Motivated by prior work such as (Li et al., 2024), which replaces attention with auxiliary MLPs to denoise each image block independently, we ablate this setting by modifying the attention mask to restrict causal attention

| FID ($\downarrow$) | Full Design | MLP | MLP (w/o Clean Blocks) |
|---|---|---|---|
| **FFHQ** | **17.8** | 21.2 | 19.8 |
| **ImageNet** | **30.0** | 96.5 | 99.9 |

to within-block scope, effectively severing information flow across blocks during the diffusion phase. This mimics independent blockwise MLP denoising. As shown in Table 5, this leads to significant degradation in generative quality, with FID scores worsening from 17.8 to 21.2 on FFHQ and from 30.0 to 96.5 on ImageNet. These results underscore the critical role of sequence-level causal attention

in facilitating coherent refinement across blocks, which we hypothesize is crucial for preserving spatial consistency and fine-grained details during generation.

## 4.6 LOSS FUNCTION DESIGN

**Auxiliary losses improve training dynamics.** To promote structured latent representations and facilitate information flow, we introduce two auxiliary losses: a *hidden loss* on AR condition latents and a *clean tower loss* on clean block latents, both supervised by shifted clean block targets (see Section 3.2). As shown in Table 6, the hidden loss notably improves FID, reducing it from 19.4 to 17.8 with a coefficient of 0.1. We hypothesize that a small weight regularizes the AR prior, whereas a large weight over-constrains it and interferes with the denoising objective. The clean tower loss has a rather diminished impact on the final generation quality. However, we observe that empirically both losses help accelerate convergence at the beginning of the training, exploring adaptive loss weighting is left for future work.

Table 6: Ablation on loss function

| **FID** ($\downarrow$) | Hidden Lambda | | | Clean Tower Lambda | | |
|---|---|---|---|---|---|---|
| | $\lambda = 0$ | $\lambda = 0.1$ | $\lambda = 1$ | $\lambda = 0$ | $\lambda = 0.1$ | $\lambda = 1$ |
| **FFHQ** | 19.4 | **17.8** | 18.4 | **17.76** | 17.79 | 18.42 |
| **ImageNet** | 30.2 | **30.0** | 31.5 | 30.04 | 30.05 | **29.98** |

## 5 RELATED WORKS

**Latent-space diffusion methods.** Diffusion models are a cornerstone of modern image generation, with early methods like DDPMs (Ho et al., 2020) operating directly in pixel space through iterative denoising, which is computationally expensive, especially at high resolutions. To address this, a trend has emerged toward latent-space diffusion. Latent Diffusion Models (Rombach et al., 2021) compress images into low-dimensional representations using a VAE, enabling efficient diffusion while preserving image quality. Expanding on this, DiT (Peebles & Xie, 2022) replaces U-Nets with Vision Transformers to scale diffusion using self-attention in latent space. Conditioning mechanisms also evolved, with DALL·E 2 (Ramesh et al., 2022) and Imagen (Saharia et al., 2022) leveraging pretrained text encoders like CLIP and T5 to guide generation with strong semantic grounding. These innovations have enabled scalable and photorealistic image synthesis that serves as a key component in broader multimodal systems.

**Unified multimodal frameworks.** As multimodal generation becomes increasingly important, a central challenge is how to model text and vision jointly with a unified architecture. Recent frameworks converge on using a single Transformer to handle both modalities in an interleaved fashion. For example, Transfusion (Zhou et al., 2024) and Show-o (Xie et al., 2024) combine AR text modeling with diffusion-based image generation, sharing token sequences and switching objectives midstream. Others build upon frozen language models for extensibility: LMFusion (Shi et al., 2024) inserts parallel diffusion layers into a frozen Llama-3, while MonoFormer (Zhao et al., 2024) toggles attention masks to support both causal and bidirectional flows. LatentLM (Sun et al., 2024) proposes encoding continuous modalities with a VAE and performing AR generation using next-token diffusion, bridging discrete and continuous domains. To further scale, MoT (Liang et al., 2024) introduces sparse modality-specific routing to efficiently pretrain over diverse input types. Unified-IO and Unified-IO 2 push this idea further by tokenizing diverse modalities into a shared vocabulary (Lu et al., 2022c; 2023). Recent advancements like GPT-4o embed generative image capabilities natively within an omnimodal architecture, enabling seamless and expressive image-text interaction (OpenAI, 2025). Together, these works represent a broader shift from modality-specific pipelines toward unified generative models.

**Hybrid AR and diffusion architectures for image generation.** While unified frameworks often separate processing text and vision, an orthogonal line of research blends AR and diffusion modeling for image generation itself. This hybridization aims to combine the sampling efficiency of AR with the

fine-grained flexibility of diffusion. For instance, Visual Autoregressive Modeling (Tian et al., 2024) generates images from coarse to fine resolutions, achieving strong performance by modeling images hierarchically. HART (Tang et al., 2024) introduces a hybrid tokenizer that factors VAEs into discrete AR tokens for global structure and continuous latents for local details, refined via a lightweight diffusion module. ACDiT (Hu et al., 2024) takes a more granular approach by interleaving AR and diffusion steps at the block level. Block Diffusion (Arriola et al., 2025) autoregresses over blocks while running discrete diffusion within each block—interpolating between AR and diffusion, enabling variable-length generation and parallel token sampling—conceptually aligned with our hybrid design. Other efforts (Li et al., 2024) and (Tschannen et al., 2023) explore replacing vector-quantized tokenizers with continuous AR modeling via diffusion-based losses, or real-valued latent sequence modeling with Gaussian mixture outputs, achieving strong generation results while bypassing the limitations of discrete token spaces. These hybrid models suggest that future image generators may increasingly blur the line between AR and diffusion, dynamically adapting sampling strategies for quality, speed, or resolution requirements.

## 6 CONCLUSION

We present *MADFormer*, a unified Transformer architecture that flexibly combines AR and diffusion modeling within the image generation pipeline. By partitioning images into spatial blocks and allocating AR and diffusion layers across model depth, *MADFormer* enables structured global conditioning and high-fidelity local refinement. Through a suite of controlled experiments, we demonstrate that both block-wise and layer-wise AR-diffusion mixing contribute to improving quality and efficiency under different generation settings. Our analysis highlights the trade-offs between AR and diffusion components, offering practical guidelines for balancing model capacity under different compute budgets—in particular, having more AR layers benefits the setups with low inference compute. We hope *MADFormer* serves as a concrete step for future research into hybrid generative architectures that adaptively leverage the strengths of both AR and diffusion paradigms.

**Future Work.** We acknowledge the limitations imposed by our study's scope (constrained compute and standard sampling), and we believe that promising *future research directions* include (i) systematically mapping how the training budget repositions the AR/diffusion optimum across NFE regimes, (ii) integrating advanced sampling techniques with our capacity-allocation guidance, and (iii) extending evaluation to text-to-image (T2I) and out-of-distribution (OOD) compositionality under the same constrained compute setting.

### ACKNOWLEDGMENTS

We would like to thank Shengding Hu and Jinyi Hu for their insightful advice on research questions and implementation details. We are also grateful to Ning Ding, Yingfa Chen, Xingyu Shen, Bohan Lyu, Xinyu Wang, Gonghao Zhang and Dunjie Lu for their valuable feedback and suggestions on early drafts of this work. We appreciate the overall support from the THUNLP group.

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

## A  LLM Usage Disclosure

We used large language models (LLMs) solely for language polishing and proofreading of the manuscript text (grammar, phrasing, and minor stylistic edits). LLMs were *not* used for research ideation, experimental design, data collection, model implementation, analysis, or result generation. No confidential submission materials were provided to any LLM. All content was verified and edited by the authors, who take full responsibility for the paper's contents. LLMs are not authors.

## B  Implementation Details

Our model configurations are summarized in Table 7. Each model used in the ablation studies was trained for 256 GPU hours on NVIDIA A100 GPUs. We conducted 20 training runs each for the FFHQ-1024 and ImageNet ablation experiments.

For evaluation, we used a generation batch size of 16 for FFHQ-1024 and 256 for ImageNet.

| Parameter | Value |
|---|---|
| Architecture | LlamaForCausalLM |
| Hidden size | 1024 |
| Intermediate size | 4096 |
| # layers | 24 |
| # attention heads | 16 |
| # KV heads | 4 |
| Attention dropout | 0.0 |
| Activation function | SiLU |
| Max position embeddings | 2048 |
| RMS norm epsilon | 1e-5 |
| RoPE theta | 10000 |
| Initializer range | 0.02 |
| Vocab size | 128256 |
| **Diffusion Parameters** | |
| # diffusion layers | 24 |
| Tokens per patch | 256 |
| Train timesteps | 1000 |
| Inference timesteps | 250 |
| Beta schedule | linear $(0.0001 \rightarrow 0.02)$ |
| Prediction type | sample |
| **Loss Weights and Aux** | |
| $\lambda_{image}$ | 5 |
| $\lambda_{hidden}$ | 0.1 |
| $\lambda_{clean}$ | 0.0 |
| **UNet Architecture** | |
| Image size | 256 |
| UNet channels | [512, 1024] |
| Layers per block | [2, 2] |
| Transformer layers per block | [0, 1] |
| UNet heads | 8 |
| ResNet groups | 8 |
| Time embedding dim | 128 |

Table 7: Model configuration for *MADFormer*. Diffusion and UNet components are integrated into the Transformer backbone.

## C  ADDITIONAL QUALITATIVE RESULTS

**Additional qualitative results of *MADFormer* with 14 diffusion layers (out of 28 total).** Thanks to the AR conditioning mechanism, *MADFormer* achieves high sample quality even with limited inference steps. The model effectively captures class semantics and produces coherent image structures, though some fine details may be missing due to limited training. Increasing the number of inference steps significantly improves generation quality, especially in modeling facial details and refining backgrounds.

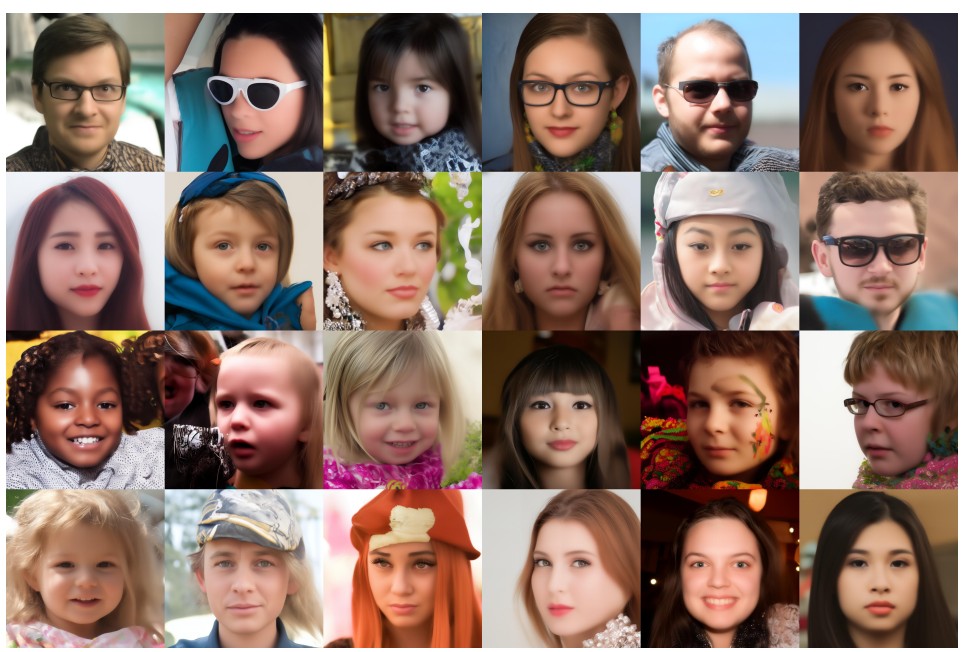

Figure 5: **Samples on FFHQ** $1024 \times 1024$ **with 9 inference steps.**

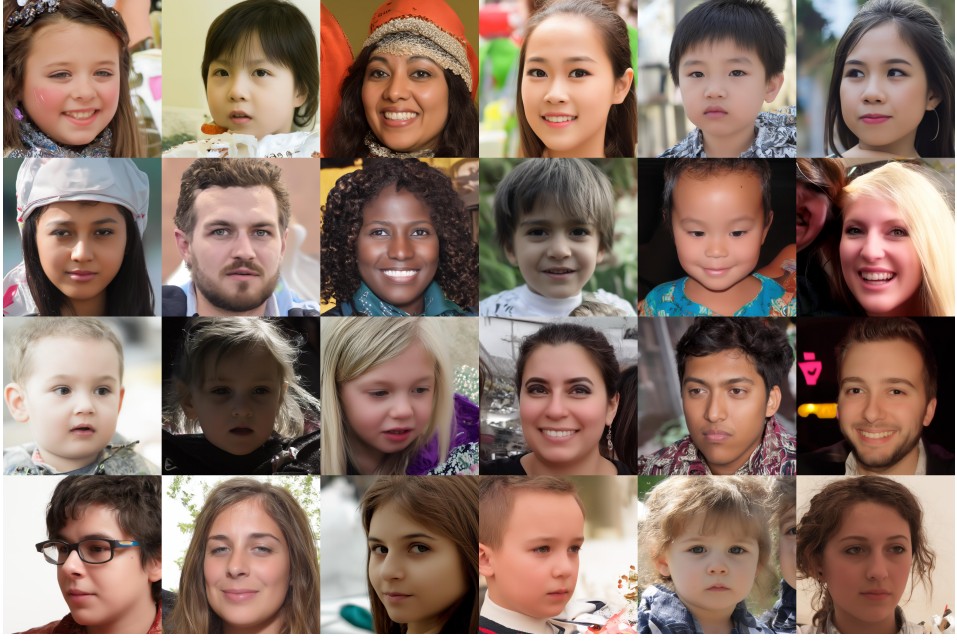

Figure 6: **Samples on FFHQ** $1024 \times 1024$ **with 19 inference steps.**

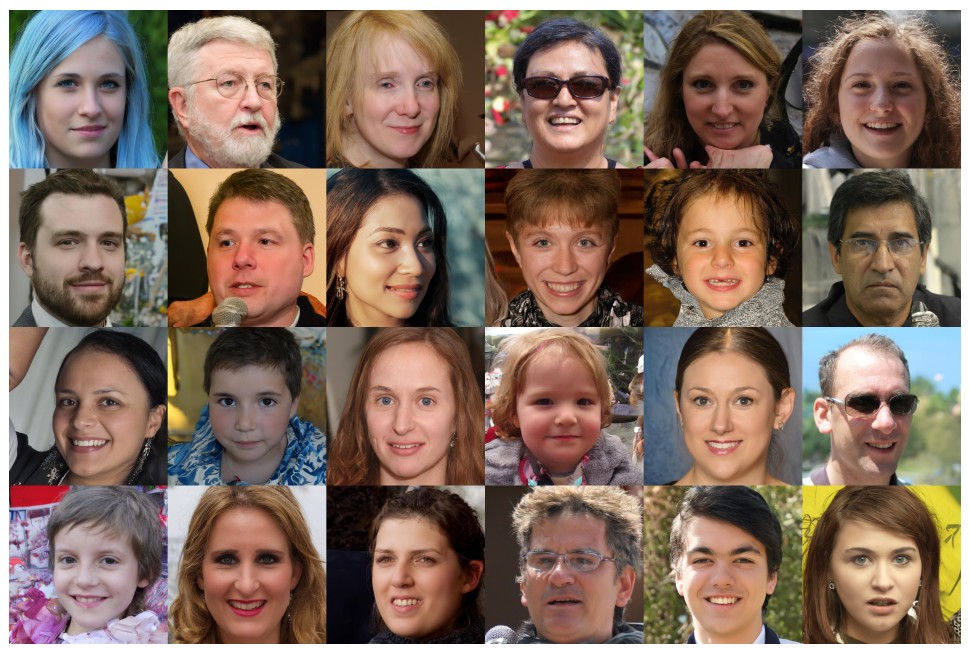

Figure 7: **Samples on FFHQ** $1024 \times 1024$ **with 49 inference steps.**

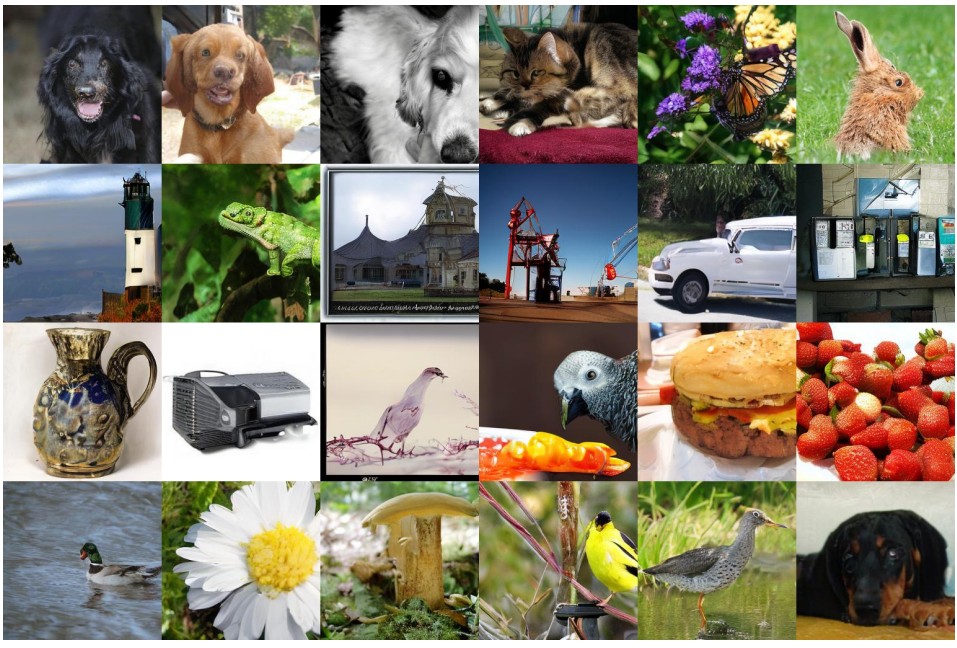

Figure 8: **Samples on ImageNet with 199 inference steps.**

