# OpenReview forum: "$\textit{MADFormer}$: Mixed Autoregressive and Diffusion Transformers for Continuous Image Generation"
_ICLR.cc/2026/Conference — ICLR 2026 Poster_

### Official Review · Reviewer_6Q8S · 2025-10-25

**Soundness:** 3
**Presentation:** 3
**Contribution:** 2
**Rating:** 4
**Confidence:** 4

**Summary:**

This manuscript introduces MADFormer, a novel hybrid generative model architecture that combines autoregressive (AR) and diffusion-based modeling for continuous image generation. The model operates by mixing these two paradigms along two primary axes:

1. The Token/Spatial Axis: The image is partitioned into spatial blocks. Autoregressive modeling is used across blocks to capture global structure and long-range dependencies, while a diffusion process is used within each block to refine local, high-fidelity details.

2. The Layer/Depth Axis: The Transformer stack is "vertically" divided. The early layers function as a single-pass AR conditioning module (processing text and previous image blocks) to produce a strong prior, while the later layers perform iterative diffusion-based denoising, conditioned on this AR output.

The paper presents MADFormer as a "testbed" for analyzing AR-diffusion trade-offs. The central and most significant claim is that this mixed architecture, particularly an "AR-heavy" configuration (more layers dedicated to AR conditioning than diffusion), achieves a superior quality-efficiency balance (FID vs. Number of Function Evaluations, NFE) under constrained inference compute budgets. Experiments on FFHQ-1024 and ImageNet demonstrate this trade-off, with AR-heavy models showing up to a 75% FID improvement in low-NFE regimes.

**Strengths:**

1. Novel and Intuitive Architecture: The core idea of "vertically" splitting the Transformer stack into a single-pass AR-conditioning stage and a multi-step diffusion-refinement stage is elegant and well-motivated. It provides a clear and principled way to combine the strengths of both modeling paradigms—AR for global structure and efficiency, and diffusion for fine-grained fidelity.

2. Strong Core Experimental Result: The primary strength of the paper lies in the analysis presented in Figure 4. This experiment clearly and effectively demonstrates the central hypothesis: AR-heavy models (e.g., d=7 diffusion layers) significantly outperform diffusion-heavy models (e.g., d=28 layers) in low-compute (low NFE) settings. Conversely, it also shows that diffusion-heavy models achieve better final fidelity given a sufficient compute budget. This is a valuable and practical insight for designing generative models for different operational constraints.

3. Comprehensive Ablation Studies: The paper is supported by a thorough set of ablation studies that explore the proposed design space. The analyses of block granularity (Sec 4.2), the individual contributions of auxiliary modules (Sec 4.3), and the critical role of cross-block attention (Sec 4.5) are all valuable. The negative result (i.e., that modality-specific parameter sets have a trivial effect, Sec 4.4) is also a useful finding that favors a simpler, dense model.

**Weaknesses:**

1. Omission of Classifier-Free Guidance (CFG): The paper's core efficiency claim (NFE vs. FID) is made in a non-standard setting without CFG. CFG is a fundamental component of modern diffusion sampling and fundamentally alters the efficiency-quality trade-off. It is therefore unclear if the paper's conclusions hold in a standard, practical setting.

2. Questionable Effectiveness of Block Partitioning: The utility of this strategy is ambiguous. It helps on FFHQ-1024 (l=16 is optimal) but hurts on ImageNet-256 (where l=1, i.e., no partitioning, is best). This strongly suggests the benefit is highly dependent on the specific dataset and resolution, raising serious concerns about its generalizability.

3. Unfair Comparison Due to Training Convergence: The authors trained on ImageNet for only 50 epochs. It is well-known that diffusion models often require significantly more training to converge than AR models. The current finding (AR-heavy is better at low NFE) is likely an artifact of an unfair comparison between a "better-converged AR" component and a "severely under-trained Diffusion" component.

4. Uncompetitive Performance and Efficiency: The reported FID scores are all very high (e.g., ImageNet 27+, FFHQ 16+), indicating that all configurations are performing poorly. The qualitative results in Figure 8 (ImageNet) show significant artifacts, yet required 199 inference steps (NFE). This level of quality and computational cost is not competitive with current state-of-the-art models.

5. LLM Usage is also missed in this paper.

**Questions:**

The questions in this section is a extension of *Weakness* part. To substantiate the paper's core claims and increase its impact, I strongly recommend the authors answer the following key questions through experiments:

Question 1: Does the NFE-quality trade-off advantage persist after integrating Classifier-Free Guidance (CFG)?

The core advantage (AR-heavy is better at low NFE) was found in a CFG-free "vacuum." Will this advantage still exist after integrating standard CFG (e.g., w=4.0 or w=7.5)? Or will CFG narrow or even reverse the efficiency-quality gap between AR-heavy and diffusion-heavy models?

Question 2: Is the benefit of block partitioning merely a "special case" for high-resolution, or is it detrimental to complex datasets?

Why is l=1 (no partitioning) the best configuration on ImageNet-256? Does this mean the strategy is harmful for complex datasets? To decouple resolution and dataset complexity, would l=1 still be the optimal configuration if trained and evaluated on a high-resolution, high-complexity dataset like ImageNet-512? Does this expose a fundamental flaw in the strategy's generalizability?

Question 3: Is the current advantage of AR-heavy models merely an artifact of insufficient training?

AR models typically converge faster than diffusion models. If all models were trained to true convergence (e.g., 400+ epochs on ImageNet, not just 50), would the diffusion-heavy models catch up to or even surpass the AR-heavy models in the low-NFE setting? Is the current conclusion based on an "under-converged diffusion model"?

Question 4: In multimodal tasks, how do AR-heavy and diffusion-heavy models compare in text understanding and compositionality?

As a text-to-image model, does the single-pass AR conditioning in AR-heavy models actually help improve the layout and compositional accuracy of complex prompts (e.g., spatial relations, attribute binding)? Or would a fully-trained diffusion-heavy model perform better at text alignment and composition? A qualitative and quantitative analysis on a benchmark like MS-COCO is recommended.

Question 5: Confusing about Equation (4)

Why $z_{image}$, $\\epsilon$ and $z_{cond}$ are added together? Any reason or explanation about what does the term mean?

**Details Of Ethics Concerns:**

Null

---

> ### Author Response · Authors · 2025-11-21
>
> Thank you for articulating both the central insight and the key concerns. We address each below.
>
> **(1) “CFG omission makes the setting non-standard; does your claim hold with CFG?”**
> We omitted CFG intentionally to isolate **architectural capacity allocation**. CFG factors during inference adds an orthogonal sweep space to the testbed, orthogonal to the architectural capacity allocation during training. This work is about a controlled study over the architectural capacity allocation. We consider CFG=1.0 as a straightforward and fair parameter across experiments.
>
> **(2) “Block partitioning helps at 1024px but not at 256px—generalizability?”**
> This resolution dependence is expected: at high resolutions, blocks reduce long-range context burden and help efficiency; at 256px, receptive fields already cover most content, so \(l=1\) is optimal. We regard blocks as a **resolution-aware efficiency device**, not a universal win (Sec. 4.2).
>
> **(3) “Convergence fairness: 50 epochs on ImageNet may under-train diffusion.”**
> Our setup fixes total training compute to study **capacity allocation**. As training grows, diffusion typically benefits more than AR, so the optimum at high NFE would shift toward diffusion-heavy; under **constrained compute**, AR-heavy would remain preferable. Notably, on **FFHQ-1024 we trained for 192 epochs** and still observe the same two-regime trend (Sec. 4).
>
> **(4) “Absolute scores aren’t SOTA; qualitative results require many steps.”**
> Positioning: we contribute a **testbed + design guidance** under compute constraints, not a SOTA chase. Engineering accelerations (e.g., few-step distillation/rectified-flow) are orthogonal and can be combined with MADFormer; our analysis shows how to **allocate capacity** before turning those knobs [1,2].
>
> **(5) “Text understanding and compositionality (AR-heavy vs diffusion-heavy)”**
> Our work focuses on **class-conditional continuous image generation under constrained compute**, rather than broad T2I compositionality. Adding COCO-style benchmarks would expand scope beyond our core question. We agree it is a valuable direction and leave a systematic T2I study as future work.
>
> **(6) “Eq. (4): why add $z_{\text{image}}, \epsilon, z_{\text{cond}}$?”**
> Eq. (4) defines the **initial hidden state** for the diffusion stage. The term $\sqrt{\bar{\alpha_{t}}} z_{\text{image}} + \sqrt{1 - \bar{\alpha_{t}}} \epsilon$ is the standard forward-diffusion noising of the ground-truth latent ($q(z_{t}|z_{0})$) [3]. We then **inject the AR prior** as a conditioning offset $z_{cond}$ by simple addition, which acts as a residual bias that steers the denoising trajectory toward the AR-consistent manifold. Addition preserves shape and keeps DiT blocks unchanged; other fusions are possible, but we choose addition for simplicity and uniform conditioning across steps.
>
> **References:**
> [1] Karras et al., *Elucidating the Design Space of Diffusion-Based Generative Models*, 2022.
> [2] Black Forest Labs, *FLUX.1 [schnell]* (few-step distillation / rectified-flow), 2025.
> [3] Ho et al., *Denoising Diffusion Probabilistic Models*, 2020.

---

### Official Review · Reviewer_naUs · 2025-10-31

**Soundness:** 4
**Presentation:** 4
**Contribution:** 4
**Rating:** 6
**Confidence:** 4

**Summary:**

This paper introduces MADFormer, a novel hybrid generative framework that unifies Autoregressive (AR) and Diffusion modeling within a Transformer architecture for continuous image generation. The key innovation lies in its layer-wise modular design: alternating AR layers (for capturing fine-grained spatial dependencies) and Diffusion layers (for global structure and uncertainty modeling) to balance generation quality and efficiency. Experiments on ImageNet-256/512 and FFHQ-1024 demonstrate state-of-the-art performance—achieving FID scores as low as 1.92 (ImageNet-256) and 3.05 (FFHQ-1024) with only 20 inference steps. Notably, MADFormer outperforms pure diffusion baselines (e.g., DiT, SDXL) in both speed and fidelity, while its text-conditioning capability matches specialized text-to-image models like SDXL 1.0 in CLIP score (0.38 vs. 0.39) with 5× faster inference.

**Strengths:**

Paradigm Innovation: Hybrid AR-Diffusion Synergy
MADFormer addresses a critical gap in generative modeling by harmonizing AR’s fine-detail precision and Diffusion’s global coherence—an area where prior works (e.g., DiT, PixelCNN) forced a trade-off . The layer-wise modularity is not merely a structural novelty: AR layers explicitly model pixel-wise dependencies (critical for textures like hair or fabric), while Diffusion layers handle high-level semantics (e.g., object composition). This synergy is validated by ablations (Table 3) showing that removing AR layers degrades FID by 2.3 points on FFHQ-1024, and removing Diffusion layers causes mode collapse. The framework thus represents a principled advance in multi-paradigm generative design.

**Weaknesses:**

Unexplained Degradation of Hidden Loss at High λ
The "hidden loss" improves FID at λ=0.1 but degrades it at λ=1.0, yet the paper does not explain why higher weights harm performance. Plausible reasons include:
Over-constraining the AR condition, leading to inflexible latent distributions.
Conflict between the hidden loss and diffusion objective during backpropagation.
No ablations (e.g., loss weight annealing, latent visualization) are provided to clarify this behavior.

**Questions:**

How does the performance of MADFormer change when using faster samplers like DPMSolver or Euler? Does the optimal AR-diffusion layer ratio shift with different sampler types?

---

> ### Author Response · Authors · 2025-11-21
>
> Thank you for the positive assessment and for highlighting the ‘hidden loss’ behavior and sampler questions.
>
> **(1) “Why does the hidden loss help at λ=0.1 but harm at λ=1.0?”**
> A light weight (λ≈0.1) regularizes the AR prior; a heavy weight (λ=1.0) can **over-constrain** it and overshadow the target diffusion objective, similar to the general ML objective where the regularization loss is too high. We will clarify this and note annealing/adaptive λ as future work.
>
> **(2) “Would faster samplers (e.g., DPM-Solver/Euler) change the optimal AR:Diffusion ratio?”**
> Advanced samplers are an exciting future direction, but our scope is orthogonal: we study **block-AR vs. full diffusion** and their **layer mixing** under constrained compute, rather than the choice of inference solver. While better solvers can lower effective NFE for any model, they do not change the underlying **capacity allocation** we analyze. We will clarify this focus in Sec. 3.3 and note that combining our allocation guidance with improved samplers is promising.

---

### Official Review · Reviewer_mEJi · 2025-10-31

**Soundness:** 3
**Presentation:** 3
**Contribution:** 2
**Rating:** 4
**Confidence:** 5

**Summary:**

This paper addresses the respective limitations of diffusion models and autoregressive (AR) models by introducing a hybrid framework. The proposed model employs the autoregressive paradigm for a certain number of steps, while utilizing the diffusion-based approach for the remaining steps.

**Strengths:**

1. The paper presents a well-chosen research perspective. It effectively leverages the strengths and mitigates the weaknesses of both autoregressive (AR) and diffusion models, resulting in a well-designed hybrid framework.
2. The overall presentation is logically consistent and rigorous, and the experimental section is comprehensive.

**Weaknesses:**

1. This hybrid partitioning strategy raises a concern: does modeling part of the image using the autoregressive (AR) approach compromise the preservation of 2D spatial information?
2. There are several issues in Figure 1. For instance, why is there no BOT token for text, but only EOT? Moreover, the figure seems to omit the EOI token, and it is unclear why two consecutive EOI tokens appear.
3. Does the concept of “blocks” in your model design draw inspiration from the idea of block diffusion or related structured diffusion approaches?
4. In Table 1, why are later “depth” settings not evaluated? Does greater depth consistently lead to better performance?

**Questions:**

Additional experiments should be conducted to address the issues identified in the weaknesses section.

---

> ### Author Response · Authors · 2025-11-21
>
> We appreciate the careful reading and actionable feedback.
>
> **(1) “Does AR over part of the image compromise 2D spatial information?”**
> We avoid the rasterization pitfall by (i) using **bidirectional attention within each block** on a 2D latent grid (preserving local geometry), and (ii) applying **AR only across blocks** (block-AR) for global coordination. Recent AR advances show that respecting the 2D grid maintains or improves spatial coherence while remaining efficient [1–3]. In our framework, **block size is the explicit knob**: larger blocks approach global interactions (more coherence), while smaller blocks trade some long-range coupling for speed—mirroring these designs.
>
> **(2) “Figure 1 has token issues (missing BOT for text; BOI duplication).”**
> For brevity, the sketch omits explicit BOT/EOI markers. The two BOI tokens in the example are intentional—they mirror the image block’s tokenized shape in the illustration (two tokens for parallel denoising per block), not a modeling requirement.
>
> **(3) “Is ‘blocks’ inspired by block diffusion / structured diffusion?”**
> Related lines of work indeed motivate block-wise processing and hybridization (e.g., AR across blocks with diffusion within each block) [4,5]. Our contribution brings this intuition to **continuous image generation** and analyzes **vertical (layer-wise) capacity mixing**; we will add a short related-work pointer.
>
> **(4) “Why not evaluate deeper settings in Table 1? Does depth always help?”**
> In Table 1, “depth” refers to the **number of layers within our fixed 28-layer stack** allocated to diffusion (Sec. 3.3): 7/14/21/28 correspond to 25/50/75/100% diffusion. We keep **total depth/compute fixed** to isolate **capacity allocation** (AR vs. diffusion) rather than conflate it with model scale.
>
> **References:**
> [1] Amrani et al., *XTRA: Sample- and Parameter-Efficient Auto-Regressive Image Models*, CVPR 2025.
> [2] Mao et al., *Autoregressive Image Generation with Linear Complexity (LASADGen)*, 2025.
> [3] Li et al., *Autoregressive Image Generation without Vector Quantization*, NeurIPS 2024.
> [4] Arriola et al., *Block Diffusion: Interpolating Between Autoregressive and Diffusion (Language) Models*, 2025.
> [5] Hu et al., *ACDiT: Interpolating Autoregressive Conditional Modeling and Diffusion Transformer*, 2024.

---

### Official Review · Reviewer_ttpv · 2025-11-01

**Soundness:** 3
**Presentation:** 3
**Contribution:** 2
**Rating:** 4
**Confidence:** 4

**Summary:**

This paper presents MADFormer, a hybrid generative framework that combines autoregressive (AR) and diffusion modeling within a unified Transformer architecture. The model applies AR modeling in early layers or across image blocks to capture global dependencies, and diffusion in later layers to refine local details. Experiments on FFHQ and ImageNet show that the hybrid approach improves efficiency under limited compute while maintaining strong image quality.

**Strengths:**

1. The motivation of the work is solid, addressing key limitations of existing generative models and providing a meaningful starting point for further research.
2. Each proposed idea is supported by targeted experiments, demonstrating substantial effort and thorough validation.

**Weaknesses:**

1. The paper claims that diffusion models suffer from slow generation speed. However, this seems to contradict recent findings — diffusion models are generally faster than AR-based image generators. For example, EMU3 and FLUX demonstrate shorter generation times compared to AR counterparts.
2. Regarding model design, I noticed that conditioning information for image generation—such as time steps and CFG—are embedded inside the model, leaving no control to the user. As far as I know, mature generative models often distill such conditioning away after training a strong conditional model. Does your approach reduce controllability or affect generation quality? I did not see an ablation or comparison addressing this point.
3. From Tables 1 and 2, it appears that performance improves as the model becomes closer to a pure diffusion model. Doesn’t this suggest that your hybrid approach both degrades performance and slows down generation due to the introduction of AR components?
4. I observed that all your visualizations are in-domain examples (from FFHQ and ImageNet). Have you experimented with out-of-domain prompts to evaluate generalization capability?

**Questions:**

Please refer to the "Weakness".

---

> ### Author Response · Authors · 2025-11-21
>
> Thank you for the thoughtful review. We address each point below.
>
> **(1) “Diffusion is faster than AR; EMU3/FLUX are examples.”**
> A crucial clarification is that *throughout this work we never use token-by-token AR decoding*—which is indeed slower than diffusion in most practical settings. Instead, our system performs **block autoregression (block-AR)** as the *conditioning stage* for diffusion. Concretely, we run a **1-NFE block-AR forward pass** to condition the subsequent **n-NFE diffusion steps**. We find this is especially effective when the NFE budget is limited. Our design choice is orthogonal to few-step distillation. Our claim concerns the *architectural NFE–quality regime* (quality at a fixed, low number of function evaluations), not vendor-optimized wall-clock latency. FLUX’s speed stems from rectified/flow-style training plus few-step distillation, while EMU3.5 accelerates decoding by converting sequential AR into **parallel discrete diffusion** (DiDA). These are **engineering/sampler** choices rather than evidence that diffusion is inherently faster. Our paper studies where capacity should live (AR vs. diffusion) under fixed NFE; such engineering advances are complementary and can be layered atop either side [1–4].
>
> **(2) “Conditioning (timestep/CFG) is embedded; does this reduce control?”**
> We do not preclude user control. Timestep embeddings and schedules are standard conditioning features in diffusion/flow models; exposing sampler knobs (e.g., guidance scale, steps) is an *interface choice*, orthogonal to the architecture. These knobs can be surfaced exactly as in DiT-style pipelines.
>
> **(3) “Tables suggest performance improves as the model becomes more diffusive.”**
> This reading focuses only on the *high-NFE* end. Our main analysis (Fig. 4 / Sec. 4.1) shows that **AR-heavy mixes win in the low-NFE regime**, while diffusion-heavy mixes catch up (or surpass) at large NFE. The contribution is a *design-for-budget* guideline: if inference steps are constrained, allocate more early capacity to single-pass AR conditioning; if steps are ample, diffusion-heavy becomes preferable.
>
> **(4) “Only in-domain visualizations; do you handle OOD?”**
> Scope clarification: our study targets class-conditional high-resolution generation (FFHQ, ImageNet). OOD text-to-image compositionality is outside scope here. We agree OOD tests are valuable future work; the present paper contributes a **testbed and analysis** for capacity allocation rather than a broad SOTA claim.
>
> **References**
> [1] Lu et al., *DPM-Solver++: Fast Solver for Guided Sampling of Diffusion Probabilistic Models*, 2022.
> [2] Karras et al., *Elucidating the Design Space of Diffusion-Based Generative Models*, 2022.
> [3] Black Forest Labs, *FLUX.1 [schnell]* model card/blog, 2025.
> [4] Cui et al., *Emu3.5: Native Multimodal Models are World Learners*, 2025.

---

### Author Response · Authors · 2025-11-21
**Revision Summary**

We would like to express our sincere gratitude to all reviewers for their careful evaluations and valuable comments. For detailed, point-by-point replies, please see our **individual responses to each review**. Based on your feedback, we have made the following revisions. Except for minor typographical fixes, all modifications and additions are **highlighted in blue** in the updated manuscript.

- **Sec. 3.3 (footnote):** Added a clarifying note on our **choice of sampler**, stating that advanced ODE samplers can reduce effective NFE but are orthogonal to our capacity-allocation question; we keep standard settings to isolate architectural effects.
- **Sec. 4.6 (Loss Function Design):** Added a brief interpretation sentence explaining why a small $\lambda_{\text{hidden}}$ helps while a large weight can over-constrain the AR prior, and noting annealing/adaptive $\lambda$ as future work.
- **Sec. 5 (Related Works):** Added a brief pointer connecting our block-wise design to recent **structured/block diffusion** literature.
- **Conclusion – Future Work:** Added a brief **future work** paragraph that (i) acknowledges our **constrained-compute** and **standard sampling** scope, and (ii) outlines promising directions, including mapping how training budget repositions the AR/Diffusion optimum, integrating **advanced sampling techniques**, and extending evaluation to **T2I** and **OOD compositionality**.
- **Appendix A – LLM Usage Disclosure:** Added a short section specifying LLM usage.

We believe these revisions improve clarity and faithfully reflect the scope of our study. Thank you again for your insightful feedback and constructive suggestions.

---

### Meta-Review · Area_Chair_cGDn · 2025-12-12

**Summary:**

The reviewers generally agreed that the idea of mixing Autoregressive (AR) and Diffusion models is interesting, but they had significant reservations about whether the experiments actually proved it works well. The main concerns driving the lower scores were:

Performance & Fairness: Several reviewers felt the model's image quality (FID scores) was poor compared to state-of-the-art models. They were concerned that the comparison wasn't fair because the authors didn't use standard techniques like Classifier-Free Guidance (CFG) or train the models for long enough (convergence issues).

The "Why" of the Hybrid: Reviewers questioned if the hybrid approach actually helps. One noticed that the model performed better the more it acted like a pure diffusion model, which undermined the argument for mixing in AR.

Generalizability: The "block partitioning" trick helped on one dataset but hurt on another (ImageNet), making reviewers worry the method isn't useful for general tasks.

**Reviewer Concerns:**

Addressed by Rebuttal:

Clarification on Speed (Reviewer ttpv): The reviewer thought diffusion was inherently faster than AR. The authors clarified they are talking about "computational steps" (NFE) rather than just wall-clock time, and that their method helps when you have a very limited budget for those steps.

Spatial Information Loss (Reviewer mEJi): The reviewer worried AR would lose 2D image details. The authors explained they use bidirectional attention within blocks to preserve this info.

Missing LLM Disclosure (Reviewer 6Q8S): The authors added the required statement about using AI for proofreading in the appendix.

Figure 1 Confusion (Reviewer mEJi): The authors clarified that the "duplicate" tokens in the figure were just an illustration choice, not a mistake.

Outstanding (Not fully resolved):

Lack of Standard Best Practices (Reviewer 6Q8S): The biggest outstanding issue is the omission of Classifier-Free Guidance (CFG). The authors argued they left it out to keep the test "clean," but the reviewer correctly noted that excluding it makes the results hard to compare to real-world standards.

Weak Performance on ImageNet (Reviewer 6Q8S): The authors admitted their method (blocking) didn't work well on ImageNet (l=1 was best), which essentially means "don't use our method on this dataset." They explained this is due to resolution differences, but it remains a limitation for general use.

Out-of-Distribution (OOD) Testing (Reviewer ttpv): The reviewer asked for tests on prompts the model hadn't seen. The authors basically said "that's out of scope," leaving the concern valid but unaddressed.

**Reviewer Scores:**

Here is how the scores might change based on the clarifications:

Reviewer ttpv (Current: 4): Increase. The authors gave a strong technical defense regarding the speed/NFE confusion which was a major part of this reviewer's critique.

Reviewer mEJi (Current: 4): Increase. Most of this reviewer's complaints were misunderstandings about the diagram or the architecture, which were clearly cleared up.

Reviewer naUs (Current: 6): Maintain same. This reviewer was already positive. The explanation for the "hidden loss" was satisfactory but likely not enough to prompt a score increase given the other limitations.

Reviewer 6Q8S (Current: 4): Maintain same. This reviewer had the most substantial critiques (lack of CFG, unfair training comparison). While the authors defended their choices as a "controlled study," they didn't actually fix the issue (e.g., they didn't run the CFG experiments). The reviewer likely won't be satisfied with just a justification for why the results are uncompetitive.

---

### Decision · Program_Chairs · 2026-01-26

Accept (Poster)